# Separation and detection of aqueous atmospheric aerosol mimics using supercritical fluid chromatography–mass spectrometry

Daisy N. Grace[1], Melissa B. Sebold[1,*], and Melissa M. Galloway[1]

[1]Department of Chemistry, Lafayette College, PA, USA
[*]now at: Department of Chemistry, University of Pennsylvania, Philadelphia, PA, USA

**Correspondence:** Melissa M. Galloway (gallowam@lafayette.edu)

**Abstract.** Atmospheric particles contain thousands of compounds with many different functional groups and a wide range of polarities. Typical separation methods for aqueous atmospheric systems include reverse-phase liquid chromatography or derivatization of analytes of interest followed by gas chromatography. This study introduces supercritical fluid chromatography–mass spectrometry as a separation method for the methylglyoxal – ammonium sulfate reaction mixture (a proxy for aqueous atmospheric aerosol mimics). Several column compositions, mobile phase modifiers, and column temperatures were examined to determine their effect on separation and optimum conditions for separation. Polar columns such as the Viridis UPC$^2$ TM BEH column combined with a mobile phase gradient of carbon dioxide and methanol provided the best separation of compounds in the mixture and, when coupled to an electrospray ionization tandem mass spectrometer, allowed for detection of several new masses in the methylglyoxal – ammonium sulfate reaction mixture as well as the possible identification of several isomers. This analysis method can be extended to other aqueous aerosol mimics, including the mixtures of other aldehydes or organic acids with ammonium or small amines.

*Copyright statement.* TEXT

## 1 Introduction

Secondary organic aerosol (SOA) comprises a significant portion of the total mass of atmospheric particulate matter and has been shown to impact human health and climate (Andreae and Gelencsér, 2006; Jimenez et al., 2009; Laskin et al., 2015). The composition of SOA can vary greatly; it typically contains large fractions of organic and inorganic material and water. There are an estimated 10,000 to 100,000 compounds in the atmosphere, many of which are found within the condensed phase (Goldstein and Galbally, 2007). Traditionally, SOA formation was thought to be a result of the partitioning of low volatility gas-phase reaction products into particles, but recent studies have shown that this uptake may be reversible and that subsequent reactions within the particle can lead to further SOA formation (Rossignol et al., 2014; McNeill, 2015). Within a particle, functionalization and oxidation can lead to the formation of cyclic compounds and oligomers via the reaction of aldehydes and organic acids with the inorganic compounds present (Powelson et al., 2013; Lin et al., 2015). These reactions can change the

physical and optical properties of the particle, including hygroscopicity and radiative forcing, which leads to changes in the radiative forcing of SOA (Hallquist et al., 2009). In order to understand how these changes affect radiative forcing and climate, it is necessary to first identify the compounds present and the reactions that are occurring within the particle.

The aqueous aldehyde – ammonium/amine reaction mixture is one class of reaction systems that has been shown to impact
aerosol growth (Lin et al., 2015; Hawkins et al., 2016; Aiona et al., 2017; De Haan et al., 2017). Light absorption by product mixtures generated from these systems often strongly resembles that of humic-like substances found in ambient SOA, indicating that these systems may provide insight into reactions occurring within aqueous SOA (Lin et al., 2015; Hawkins et al., 2016). Methylglyoxal reacts with ammonium through imine and aldol reactions to form compounds with a variety of polarities (see Fig. 1), including cyclic imidazoles and pyrazines and acyclic aldol condensation products (Sareen et al., 2010; Lin et al.,
2015; Hawkins et al., 2018). Oligomers are also formed via the addition of methylglyoxal to imidazoles or pyrazines, contributing to the complex light absorption of the system (Lin et al., 2015; Hawkins et al., 2018). Many studies have worked to understand the reactions occurring between methylglyoxal and ammonium and similar systems (e.g., aldehyde or organic acid with ammonium) to determine their significance for atmospheric radiative forcing and further reactions (Laskin et al., 2015; McNeill, 2015).

**Figure 1.** Methylglyoxal reacts with ammonia to form products with a variety of functional groups and polarities.

A significant challenge to the identification and quantification of atmospheric reaction mixtures is the separation of the compounds that compose them. This is due in part to the high degree of similarity between many of the compounds in solution (Noziére et al., 2015). The two most commonly used separation techniques are gas chromatography (GC) and liquid chromatography (LC). For many atmospheric samples, derivatization must be performed before GC analysis, which leads to increased specificity and identification. However, derivatization can also lead to side reactions and ambiguity in structural
identification (Noziére et al., 2015). Reverse-phase high performance LC (HPLC) and ultra performance LC (UPLC) are also commonly coupled to mass spectrometry (MS) for separation and identification of atmospheric compounds (Lin et al., 2015; Noziére et al., 2015; Aiona et al., 2017; De Haan et al., 2018; Jayarathne et al., 2018), and several studies have used these techniques to study aldehyde – ammonium/amine reaction systems (Kampf et al., 2012; Lin et al., 2015; Kampf et al., 2016; Aiona et al., 2017). Lin et al. (2015) and Aiona et al. (2017) provided comprehensive studies of chromophores found in the

methylglyoxal – ammonium sulfate system before and after photolysis using similar HPLC methods. Lin et al. (2015) found that an acetonitrile/water gradient with an SM-C18 column provided the best separation in 80 minutes at 0.2 mL min$^{-1}$. Kampf et al. (2012) analyzed a glyoxal – ammonium sulfate mixture with an acetonitrile/water gradient on an Atlantis T3 (C18) column in 60 minutes at 0.2 mL min$^{-1}$. A similar study analyzed the nitrogen-containing compounds from the reaction of small

dicarbonyls and amines on HPLC and UPLC (Kampf et al., 2016). The HPLC method utilized the same Atlantis T3 column and an acetonitrile/water gradient to separate these reaction mixtures in 19 minutes at 0.5 mL min$^{-1}$, while the UPLC method used a Hypersil Gold C18 column with an acetonitrile/water with formic acid gradient to separate the compounds in 8.5 minutes at 0.5 mL min$^{-1}$ for analysis via targeted MS/MS. The use of tandem MS coupled to both chromatography systems in that study allowed for the identification of many compounds without complete separation. While GC and LC have provided many

important insights into numerous atmospheric systems, there is a need for a separation method for aldehyde – amine reaction systems that does not require derivatization and can reduce the necessary separation time while still providing separation of a majority of the compounds in the mixture.

    Supercritical fluid chromatography (SFC) has become popular in recent years as an alternative to GC and LC in many applications (Bernal et al., 2013; Lesellier and West, 2015; Bieber et al., 2017). SFC is often thought of as analogous to

normal-phase chromatography and provides an attractive alternative to traditional LC since it is considered to be a greener technique due to the use of carbon dioxide as the main component of the mobile phase (Taylor, 2008). Less solvent waste is generated than in traditional LC, and carbon dioxide is cheap and non-toxic (Patel et al., 1998). When SFC was first developed, carbon dioxide was the only component of the mobile phase (Bernal et al., 2013). However, carbon dioxide is also miscible with many polar organic solvents as mobile phase modifiers and ion-pairing reagents as additives (Parlier et al., 1991). Incorporating

modifiers and additives can change the polarity of the mobile phase, thereby making it possible to separate a range of polar or nonpolar compounds and allowing SFC to be used analogously to either normal- or reverse-phase LC (Guiochon and Tarafder, 2011). As it is possible to use a mobile phase gradient that ranges from pure carbon dioxide to ∼50% modifier, the mobile phase can be changed from nonpolar to relatively polar over the course of one injection onto the column. This makes SFC ideal for the separation of mixtures containing compounds with a wide range of polarities, such as the methylglyoxal – ammonium

sulfate reaction system. The output from an SFC column can be coupled to a variety of instruments for analysis (Bernal et al., 2013; Bieber et al., 2017). Two commonly used mass spectrometry ionization methods are electrospray ionization (ESI) and atmospheric pressure chemical ionization (APCI). Both are soft ionization techniques that allow for identification of the molecular ion peak of a compound, which are ideal for product identification in a complex reaction system like an aqueous atmospheric reaction mimic. Many previously identified compounds from aldehyde – amine reaction systems have also been

observed using these ionization techniques (Kampf et al., 2012; Sareen et al., 2013; Lin et al., 2015; Wong et al., 2017).

    The methylglyoxal – ammonium sulfate mixture is a model system to optimize for SFC due to the fact that much is known about its chemistry and observed products (Sareen et al., 2010; De Haan et al., 2011; Sareen et al., 2013; Lin et al., 2015; Rodriguez et al., 2017; Wong et al., 2017). This reaction provides atmospherically relevant analytes to study, as well as a system that is difficult to separate since it contains many polar oligomers and reduced nitrogen compounds (see Fig. 1) (Laskin et al.,

2015; Lin et al., 2015). The ability to efficiently separate compounds in this and similar systems may allow for identification

of compounds that contribute to ambient aerosol mass. In this study, experimental conditions for five columns, four mobile phase modifiers, and a range of temperatures are evaluated and optimized to determine appropriate SFC separation conditions for this complex mixture and others like it. A method that couples SFC to ESI-MS is presented that allows for the separation and identification of products within this complex mixture in 30 minutes or less and with no sample preparation.

## 2 Materials and methods

### 2.1 Reagents

Methylglyoxal (40% w/w in $H_2O$) and ammonium sulfate were purchased from Sigma Aldrich. Food grade carbon dioxide was obtained from Airgas. Methanol (Optima[TM] LC/MS Grade), acetonitrile (Optima[TM] LC/MS Grade), ammonium formate (10 mM with 0.05% formic acid) in methanol (LC/MS grade), and formic acid (Optima[TM] LC/MS Grade) were purchased from Fisher Chemical.

### 2.2 Methylglyoxal and ammonium sulfate mixtures

Separate standard solutions of 1 M methylglyoxal and ammonium sulfate were prepared in deionized water. Mixtures for analysis were prepared by mixing sufficient volumes of each stock solution with deionized water to make solutions containing 50 mM each of methylglyoxal and ammonium sulfate. Due to slow room temperature reaction times at these concentrations, the mixture was allowed to react for 6–7 weeks in a dark environment before analysis (Zhao et al., 2015). This ensured that the reaction had proceeded far enough to form previously identified major products (Amarnath et al., 1994; Bones et al., 2010; Sareen et al., 2010; De Haan et al., 2011; Lin et al., 2015; Kampf et al., 2016; Aiona et al., 2017; Hawkins et al., 2018).

#### 2.2.1 Mass spectrometry

The methylglyoxal – ammonium sulfate mixture was separated and analyzed with a Waters ACQUITY Ultra Performance Convergence Chromatography (UPC$^2$) SFC system coupled to a Waters XEVO TQD triple quadrupole mass spectrometer. The XEVO TQD is equipped with an ESCi ion source which allows for rapid switching between ESI and APCI modes, and all samples were analyzed via positive and negative ESI and APCI modes. ESCi probe conditions were set as follows: desolvation temperature = 200°C, desolvation gas flow = 650 L hr$^{-1}$, cone flow = 1 L hr$^{-1}$. ESI conditions were set as follows: capillary voltage = 1.18 kV, cone voltage = 30 V. APCI conditions were: corona voltage = 1.5 kV, cone voltage = 50 V.

### 2.3 Supercritical fluid chromatography

Unlike an LC system, the pressure in an SFC column must be maintained throughout, so a backpressure regulator is installed after the column to ensure that the mobile phase stays in a near-supercritical state throughout the entire column. In the UPC$^2$ system, liquid carbon dioxide is pulled into a chilled carbon dioxide pump and mixed with co-solvent before delivery to the column. A 10 μL sample loop is plumbed inline and diverts solvent flow through the loop upon sample injection. Part of the

flow is sent to the backpressure regulator (set at 1500 psi), and the remaining flow is directed to the MS. In these experiments, the flow from the UPC$^2$ system was mixed with the output of an isocratic pump that provided 0.25 mL min$^{-1}$ of 10 mM formic acid in methanol as makeup flow into the ionization source.

The chromatography system was optimized using a variety of columns, modifiers, and column temperatures, as described below.

### 2.3.1 Columns

The columns used for analysis were chosen for their range of polarities and variety of functional groups. They are: AC-QUITY UPLC® BEH Amide (BEH Amide), CORTECS$^{TM}$ UPLC® HILIC (HILIC), Viridis UPC$^2$ $^{TM}$ BEH (BEH), AC-QUITY UPLC® BEH C18 (BEH C18), and Viridis UPC$^2$ $^{TM}$ BEH 2-Ethylpyridine (BEH 2-EP). Specific details about each column and chromatographic conditions can be found in Tables S1 and S2.

### 2.3.2 Modifiers

A binary gradient of carbon dioxide and organic modifier was used for elution of all samples. The total flow rate was held constant at 1.0 mL min$^{-1}$. The modifiers tested in this study were acetonitrile, methanol, 10 mM formic acid in methanol, and 10 mM ammonium formate in methanol. Optimal mobile phase conditions varied slightly with the identity of the column, but all runs started with a low percentage of modifier (0–2%), held at this concentration for 2–5 minutes, increased to 45% modifier until 15–22 minutes, then held at 45% modifier until approx. 27 minutes before returning to initial conditions for the last 3–4 minutes of the run. The initial isocratic hold was varied slightly depending on the polarity of the column since stationary phase composition significantly changed the retention of early eluting compounds. Specific details can be found in Table S2. When switching modifiers, the columns were allowed to flush with the new modifier for at least 1 hour at 1.0 mL min$^{-1}$ to ensure there were no residual additive ions on the column (Berger and Deye, 1991).

### 2.3.3 Column temperature

The temperature of the columns was varied from 35–55°C to determine the effect of temperature on separation. The optimal mobile phase conditions determined in Sect. 2.3.2 were used at all temperatures.

## 3 Results and discussion

### 3.1 Mass spectrometry

Data collection was performed with both ESI and APCI ionization modes for comparative purposes. Optimizing the MS method for each individual mass is time-consuming during both method development and data collection, so each ionization method was generally optimized to maximize as many signals as possible using direct infusion into the MS. The mobile phase in SFC separations is acidic (Lesellier and West, 2015), which helps to protonate analyte molecules during the ESI ionization process.

Therefore, it is not surprising that the overwhelming majority of these compounds are detected in positive ESI mode since they contain alcohols and nitrogen-containing functional groups that are easily protonated. The presence of formic acid in the makeup flow (ESI solvent) enhances ionization of any compound more basic than the resulting solution, making its addition very useful for these analyses. Most compounds were also detected in APCI mode but at much lower intensities than in ESI mode ($\sim 20\times$, see Fig. S12). Therefore, ESI is the preferred mode for analysis of this aerosol mimic system, though there may be some compounds that have lower ionization efficiency with ESI and may benefit from the use of APCI in some solutions, as it has often been used for analysis of slightly less polar compounds such as polycyclic aromatic hydrocarbons, esters, and pyrazine derivatives (Walgraeve et al., 2010; Laskin et al., 2015; Noziére et al., 2015; Laskin et al., 2017; Hawkins et al., 2018). As a range of polarities are found in many atmospheric samples, the ability to switch back and forth between modes in the same separation is useful for such analyses. All masses observed in this study are given in Table S3 and all extracted ion chromatograms (EICs) are shown in Fig. S14. The chromatograms presented in this study are a combination of all EIC signals in Table S3 and Fig. S14.

## 3.2 Chromatographic conditions

### 3.2.1 Columns

The packed columns used for SFC separations are similar to those used for LC systems, and many UPLC columns can be used with an SFC system. Under the conditions presented here, nonpolar compounds should elute earlier than polar compounds on a reverse-phase column since the polarity of the mobile phase increases over the course of the separation. As some compounds in this mixture are highly polar, most of the columns that were chosen for this work are intended to separate polar compounds in the slightly acidic environment (pH 4–5) present during SFC separation (see Table S1) (Lesellier and West, 2015). The BEH C18 column was chosen as a nonpolar comparison that is similar to those used in previous studies to separate imidazole derivatives and other polar molecules with SFC (Parlier et al., 1991; Patel et al., 1998; Lesellier and West, 2015). HILIC columns are commonly used to separate atmospheric compounds with LC (Noziére et al., 2015; Laskin et al., 2017) and have previously been used for the separation of samples containing a range of polarities with an SFC system (West et al., 2012; Bieber et al., 2017). Therefore, several HILIC stationary phases were chosen for this work. The HILIC column is a polar unbonded stationary phase, and the BEH stationary phase is an ethylene bridged HILIC formulation. Both are intended to separate polar compounds. The BEH Amide and BEH 2-EP columns are modified BEH columns, with amide or 2-ethylpyridine groups bonded to the stationary phase. Both columns have previously been used for the separation of polar compounds containing amines and alcohols, functional groups found in the methylglyoxal – ammonium sulfate reaction mixture (Lesellier and West, 2015).

All columns provided better separation of the compounds that eluted within 11 minutes than later eluting compounds (e.g., *m/z* 83, 97, and 126), which either coeluted or had very wide, noisy peaks that overlapped significantly depending on eluent conditions (see Figs. S1-S8). These peaks can easily be distinguished via their *m/z* values through the use of EIC. Two of these late eluting compounds have been identified as small methylimidazole derivatives (*m/z* 83 and 97), and it is likely that they

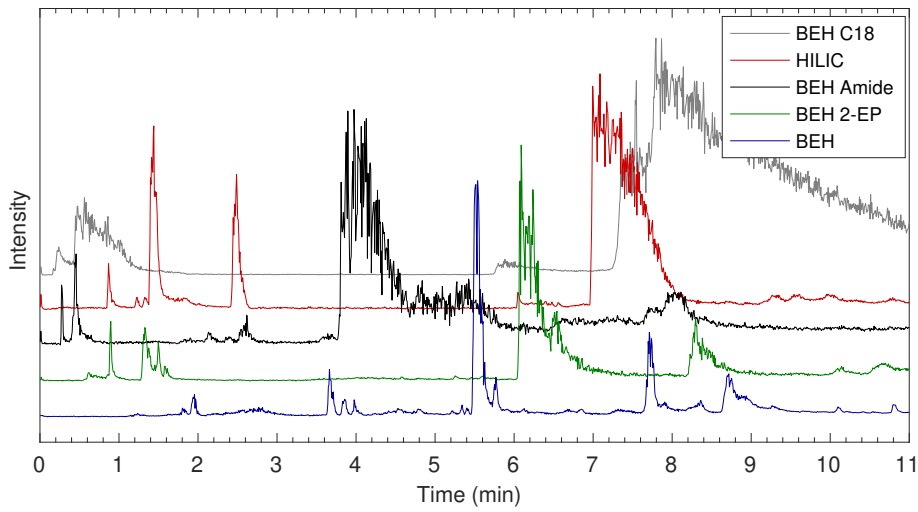

**Figure 2.** Comparison of EICs from five columns using a methanol modifier. The first 11 minutes are shown to make clear the differences in separation between lower intensity peaks. Full chromatograms are shown in Fig. S3 and the masses monitored are shown in Table S3. The more polar BEH columns (BEH, BEH Amide, BEH 2-EP) provide better separation, and the BEH provides the best separation of those columns tested.

interact very strongly with the columns and require a highly polar mobile phase for complete elution. Other methylimidazole derivatives with attached methylglyoxal oligomers elute much earlier, likely since the addition of methylglyoxal moieties decreases molecular polarity and, as a result, interactions with the column are decreased.

It was not possible to separate the compounds of interest with the BEH C18 column, as can be seen in Figs. 2 and S1-S8,
likely due to the nonpolar stationary phase. With a methanol modifier, most compounds eluted within 2 minutes even when the starting conditions contained 100% carbon dioxide. Varying degrees of separation were observed with the HILIC-based stationary phases (HILIC, BEH Amide, BEH 2-EP, and BEH). Elution times for many compounds range from <1 minute to 20 minutes, which is approximately when the mobile phase reaches its most polar condition. EICs for the first 11 minutes on each column using a methanol modifier are shown in Fig. 2, and all other chromatograms are shown in Figs. S1-S8. It was
expected that the HILIC column would efficiently separate this mixture since it is composed of bare silica and is often used in separations of similar, highly polar aqueous mixtures (Laskin et al., 2017). However, while separation on the HILIC column was improved over that of the BEH C18 column, there were still many wide, coeluting peaks between 1 and 3 minutes along with peaks that eluted much later (7 and 13 minutes), indicating that while some separation is occurring, many compounds are not well separated. BEH columns with polar functionalities such as the BEH Amide and BEH 2-EP combined with a
polar methanol modifier provided improved separation, as the methylglyoxal – ammonium sulfate mixture produces many polar compounds that contain nitrogen- and oxygen-containing functional groups that have heightened interactions with the nitrogen-containing stationary phase (amide or 2-ethylpyridine) in these columns. However, both exhibited similar features as the HILIC column, with several compounds eluting within 3 minutes, followed by wider peaks between 4 and 9 minutes. With

all modifiers tested, the best separation was achieved with the BEH column, with elution spread out until approx. 11 minutes using the methanol-based modifiers. The bridged ethylene groups on the BEH column slightly reduce the polarity of the stationary phase as compared to the open silanol sites on the HILIC column while still providing a polar stationary phase and separation of highly polar compounds. In addition to improvements in resolution while using the BEH column over the other HILIC columns, the less intense peaks that eluted before 11 minutes also had higher intensities than with the other columns. Therefore, further analysis will focus on the BEH column, although the temperature and modifier conditions discussed below were tested on each column with consistent results, as can be seen in Figs. S1-S11 in the Supplemental Information.

### 3.2.2 Modifiers

Four modifiers were tested to determine suitable mobile phase conditions for separation on each column (see Fig. 3 for an example and Figs. S1-S8 for all chromatograms). Common SFC modifiers include small alcohols (e.g., methanol and ethanol) and acetonitrile. Acetonitrile has a lower polarity index than methanol and was initially tested. In a carbon dioxide/acetonitrile gradient, compounds were still eluting from the column after 27 minutes, when the mobile phase was switched back to initial conditions (see Fig. S1). Preliminary testing showed that these compounds are not finished eluting even if the modifier is held at 45% for an additional 15 minutes, indicating that acetonitrile is not polar enough to elute all compounds present from any of the columns within 40 minutes. This is likely because a higher polarity solvent is needed to elute some of the more polar compounds from the column. There was also no improvement in separation of the earlier eluting compounds when using acetonitrile, and in most cases, separation efficiency decreased. These observations, combined with the fact that significant precipitate forms when the reaction mixture is diluted in acetonitrile in the bulk phase, led to the decision to exclusively use methanol-based mobile phases in this work. Most compounds within these mixtures are soluble in methanol, and methanol provided improved separation over acetonitrile. Methanol is the most polar of the commonly used SFC modifiers, which could explain the improved separation of the high polarity products in the methylglyoxal – ammonium sulfate reaction mixture. This study then focused on determining which mobile phase additive provided the best separation for the methylglyoxal – ammonium sulfate system.

Since SFC is often coupled to a mass spectrometer, many additives are salts that can be used as ionization agents (e.g., ammonium formate, formic acid, and tetramethylammonium hydroxide) (Cazenave-Gassiot et al., 2009; Lesellier and West, 2015). When using 10 mM ammonium formate in methanol as the mobile phase modifier, separation of compounds that elute in less than 11 minutes is similar to that of a pure methanol modifier (Fig. 3), and compounds that elute after 11 minutes do so with better resolution and with sharper peaks than the methanol or methanol with formic acid modifiers. However, the addition of ammonium in the mobile phase or makeup flow leads to artificially high signals from nitrogen-containing compounds, even in samples that contain no nitrogen (e.g., aqueous methylglyoxal). These compounds are also seen in samples containing ammonium sulfate, but their signals are enhanced with additional ammonium added into the system via the mobile phase or makeup flow and do not always elute at the same times as in these systems. Thus, the increased nitrogen-containing compounds are being formed within the instrument. Ammonium is reacting with carbonyls in the sample either on the column or in the ionization source, most likely in the ionization source. Previous studies have noted increased oligomer signals as a result of

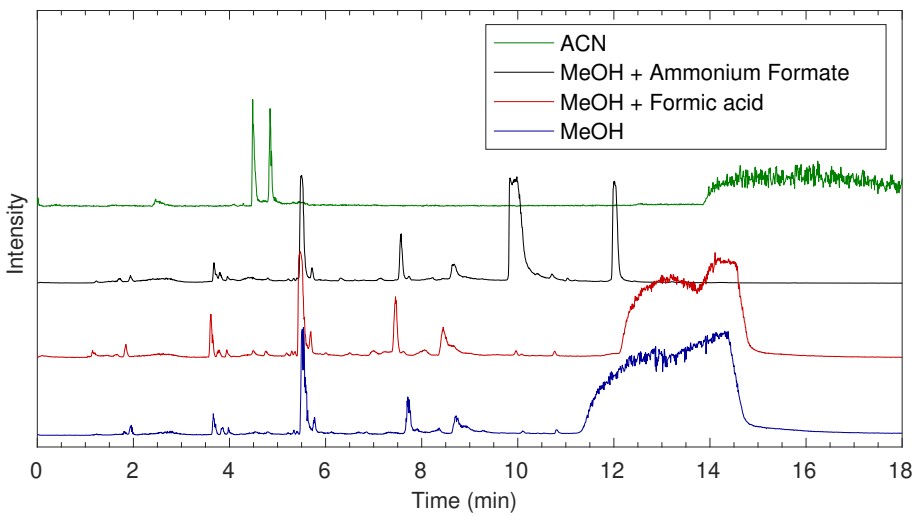

**Figure 3.** Comparison of EICs showing chromatography from methanol-based modifiers on the BEH column. The masses monitored are shown in Table S3. Mobile phase modifiers can affect separation on the column as well as chemistry occurring in the ionization source.

ESI ionization, likely due to the rapid increase in analyte concentrations within the droplets upon drying (Hastings et al., 2005). This is likely happening here, with methylglyoxal and ammonium reacting within the droplets. This is further supported by the fact that while some earlier masses elute at similar times to the methanol system, there are nitrogen-containing compounds detected at times that do not match peaks eluted with a pure methanol modifier. It is possible that compounds eluting from the column at this time are methylglyoxal oligomers formed through aldol condensation that then react with the ammonium after elution (Krizner et al., 2009). It is also possible for analytes to react with the mobile phase during SFC analysis (Lesellier and West, 2015). Therefore, the chosen additive must be one that does not react with the analytes of interest within the instrument, and additives containing ammonium are not suitable for systems containing carbonyl compounds. Since ammonium formate additives in this system lead to falsely enhanced signals of nitrogen containing compounds, no further analysis was carried out in this work, and ammonium formate modifiers are not included in Figs. S1-S7.

Formic acid is another common SFC mobile phase modifier that promotes ionization in the ESI source and does not contain the ammonium that can react with analytes in the mixture. The use of pure methanol or 10 mM formic acid in methanol as the mobile phase resulted in similar separations. Therefore, either of these modifiers could be used for separation of these compounds, and the optimal modifier will depend on the compound mixture in question. As there is little difference between separations with methanol and methanol with formic acid on the BEH column in this system, further analysis in this work uses pure methanol as the mobile phase modifier.

### 3.2.3    Column temperature

The solvating power of a super- or sub-critical fluid depends on the density of the fluid, which is affected by the temperature and pressure of the system. Therefore, SFC is similar to GC and LC in that retention and separation are not only controlled

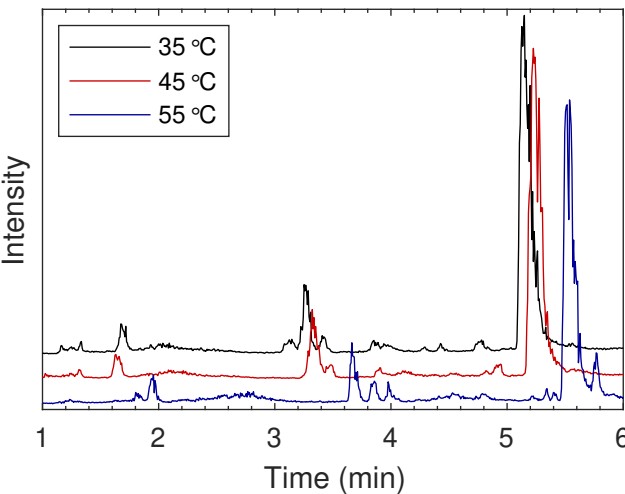

**Figure 4.** Comparison of EICs showing separation at various temperatures on the BEH column with MeOH as the modifier. Separation is improved at higher temperatures. On all columns, most peaks elute slightly earlier at lower temperatures and are somewhat sharper (see Figs. S9-S11).

by the stationary and mobile phases, but also temperature and pressure (Saito, 2013). Column temperature may become an important parameter for separation of compounds and can significantly change retention. At lower temperatures, mobile phase density and solvating power increase. When this occurs, retention times tend to shorten, which is the opposite of what would be expected for GC or LC (Saito, 2013). However, this is balanced by the effect of lower temperatures on the kinetic partitioning

of analytes into the stationary phase as is typically seen in LC or GC. Due to these several convoluting factors affecting analyte-mobile phase interactions, it is useful to test the effect of temperature on the separation of these mixtures. In this work, column temperature was varied from 35°C to 55°C to test the working range of the SFC columns. The critical temperature of carbon dioxide is 30.1°C, so pure carbon dioxide is supercritical under all of these temperature conditions, but addition of a modifier or additive to the mobile phase raises the critical temperature of the system (Guiochon and Tarafder, 2011; Saito, 2013). It is

very likely that the mobile phase is subcritical for these lower temperature analyses, but Guiochon and Tarafder (2011) showed that it does not matter whether the mobile phase is supercritical or subcritical, as long as the retention factors fall within a useful range, and analyte does not precipitate on the column during analysis.

    Changing the temperature of the column did not consistently affect separation of analytes between columns (see Figs. S9-S11 for a comparison). Retention time typically changed slightly with an increase in temperature, but this small change did

not necessitate a change in mobile phase conditions for optimal separations. No trend in retention time was seen with some modifier-column combinations, such as methanol-HILIC or methanol-BEH 2EP, while others showed obvious differences, such as acetonitrile-BEH, formic acid-BEH Amide, and methanol-BEH. Beyond retention times, peak shapes can also be affected by column temperature, as can be seen on the BEH column with an acetonitrile modifier. The peak at 6 minutes at 35°C elutes as a narrower peak in 4.8 minutes at 55°C (see Fig. S9). On the BEH column with the methanol modifier, separation

improves at 55°C as compared to 35°C or 45°C (see Fig. 4). With this system, retention times tended to increase with increased temperatures, which was generally the opposite of what was observed for the other columns. As the solvating power of the mobile phase increases, interactions between the mobile phase, stationary phase, and analyte begin to change. Interestingly, as can be seen in Fig. 4, not all analytes are affected to the same degree by this change in mobile phase. The retention time of the larger peak (mostly comprised of *m/z* 125) at 5.2 minutes at 35°C is not significantly affected as temperature is increased, but the fact that the small peak (*m/z* 181) at 5.8 minutes separates from this larger peak at 55°C indicates that this compound was far more affected by the change in temperature than the compound or compounds that comprise the larger peak. Other compounds that are affected by this change in temperature elute near 3.5 minutes and have better resolution at 55°C. Overall, higher temperatures lead to improved separation with the BEH column. For other columns, temperature did not have as large of an impact on separation, but slight improvements in peak shape were observed as temperature decreased (see Figs. S9-S11). Thus, the effects of changing column temperature on the separation efficiency of a system depend strongly on the column, mobile phase, and analyte. Therefore, temperature is a variable that must be tested for each individual system to determine how separation will be affected.

### 3.3 Comparison to LC

In order to ensure that this system performs as well as comparable chromatography – mass spectrometry coupled systems, masses detected in this system were compared to those found in the literature for the methylglyoxal – ammonium sulfate system (Amarnath et al., 1994; Bones et al., 2010; Sareen et al., 2010; De Haan et al., 2011; Lin et al., 2015; Kampf et al., 2016; Aiona et al., 2017; Hawkins et al., 2018). Each of the masses in Table S3 were monitored. While the retention times of compounds cannot be directly compared from column to column in the SFC system or between SFC and LC in order to confirm structures, the observed masses can still be compared. All previously published masses shown in Table S3 were observed as well as some that have not been seen in the literature. Figure 5 shows EICs from selected *m/z* values. Many of the observed masses elute in multiple peaks, as reported by Lin et al. (2015). EICs are useful to determine the elution time of individual masses, especially in the case of coeluting peaks, and allow for further analysis of that mass (Lin et al., 2015). Chromatograms depicted in green correspond to masses that have been shown to be important chromophores in previous work, and those in black have not yet been published for this system (Lin et al., 2015; Hawkins et al., 2018). Peak intensities for many of these new masses are comparable to those that have already been detected. Many of these peaks are also very low in intensity compared to *m/z* 83, 97, and 125 (imidazole derivatives), probably due to the variation in ionization efficiency of the ESI source for different functional groups and concentrations of products. However, many peaks are apparent when viewed as individual EICs, and it becomes possible to detect multiple compounds within the reaction mixture. Therefore, the intensities in Fig. 5 have been normalized to see separation between compounds. The highest intensity in each chromatogram is given in the right-hand column.

Through the use of tandem MS, it is possible to determine similarities between the observed molecules. Common building blocks for the methylglyoxal – ammonium sulfate reaction system include methylimidazoles. These compounds fragment to several common masses, including *m/z* 69, 83, and 97 (Kampf et al., 2016). The use of tandem MS helps to confirm the presence

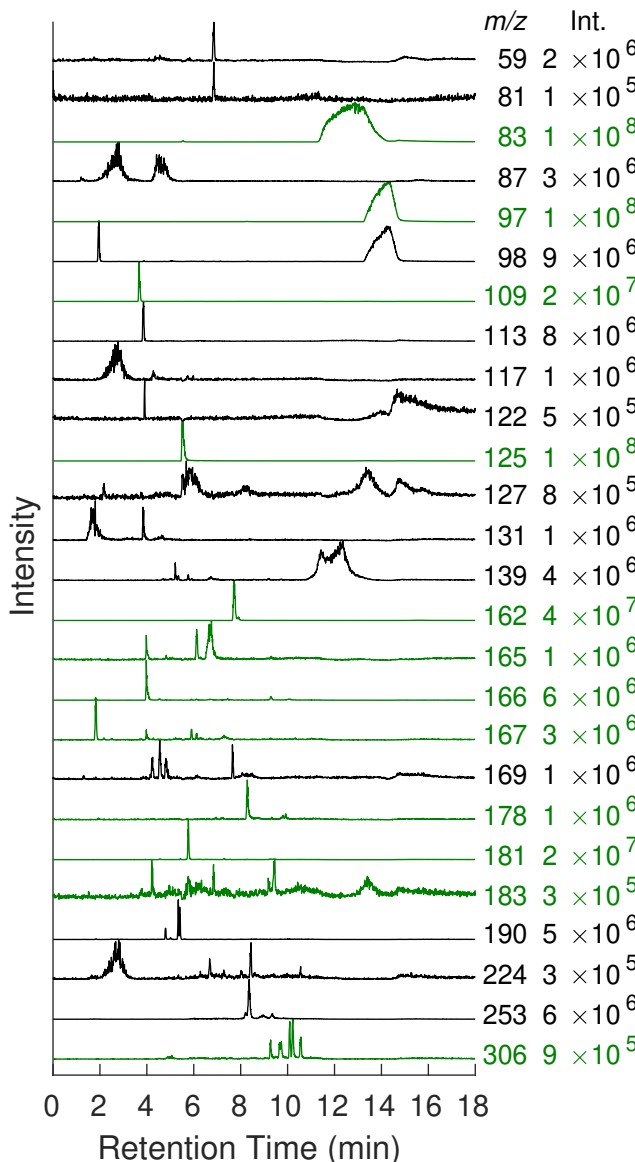

**Figure 5.** EICs of selected *m/z* values seen under the conditions described above. The intensity of the largest peak is given to the right of each trace. All chromatograms have been normalized to their maximum value for ease of viewing due to the very intense signals given by imidazole derivatives. Traces shown in green correspond to masses that have been identified in previous work. Those those in black have not yet been published for this system (Lin et al., 2015; Hawkins et al., 2018).

of methylimidazole in previously published masses such as *m/z* 125, 196, 197, 232, and 269. All of these masses contained *m/z* 83 as a fragment (see Fig. S17), indicating that these methylimidazole derivatives fragment to *m/z* 83 more readily than *m/z* 69 or 97. Therefore, *m/z* 83 was used as an identifying fragment mass for compounds that contain methylimidazole.

Several of the previously undetermined compounds in this system fragment to *m/z* 83, including *m/z* 139, 169, 190, and 253 (see Table S3 and Fig. S17). These compounds are likely to contain a substituted methylimidazole group. This conclusion is further supported by the fact that these compounds all elute within the range of retention times of previously identified methylimidazole compounds, as can be seen in Fig. 5. While it is not surprising to detect methylimidazole compounds within this system, the combination of tandem MS and chromatography that allows for separation of compounds with similar masses allows for the observation of these low intensity signals that have not been identified in previous studies.

Tandem MS also indicates that many masses that elute in multiple peaks may in fact be very similar compounds since the fragmentation patterns for multiple peaks are similar or identical. Many of the higher mass compounds with multiple retention times may be isomers with slight differences in structure due to oligomerization reactions occurring on different carbons within each molecule. This is true for masses such as *m/z* 126. Both compounds that lead to the highest intensity peaks in this EIC fragment to *m/z* 42, 55, 70, 80, 98, and 108 (see Fig. S15). Similarly, there are two major peaks in the EIC for *m/z* 165 (6.1 and 6.7 minutes, see Fig. 5), and both fragment to *m/z* 43 (very low intensity), 123, and 147 (see Fig. S16). Since these compounds elute at slightly different times, it is likely that they are isomers that reacted slightly differently as they oligomerized but have the same general structure. In general, this seems to hold true in the higher mass compounds, suggesting that differences in structure do not start to appear until higher order oligomerization has occurred (i.e., more methylglyoxal units have been added).

## 4 Conclusions

The use of SFC is ideal for separation of compounds with a wide range of polarities and is thus an excellent alternative for separating aqueous atmospheric aerosol mimic solutions containing compounds with a variety of functional groups. There are many options for columns and mobile phase modifiers that can be used to fine-tune the separation of multiple compounds of interest. For systems containing such a large range of polarities, a polar or polar functionalized column with a highly polar mobile phase such as pure methanol or methanol with an acidic additive is ideal for this separation. ESI is an attractive and commonly used ionization source for mass spectrometry and allows for detection of the compounds separated via SFC. The ability to monitor various masses via EIC or tandem MS along with a separation technique that can separate many of the less intense peaks in the chromatogram leads to the observation of previously published products of this system and several new masses, including the detection of likely isomers.

Many other small aldehydes and organic acids are found within aqueous SOA that react with ammonium and other amines to form similar product mixtures to that studied here. These have been explored by several groups (Heald et al., 2005; Hallquist et al., 2009; Jimenez et al., 2009; Laskin et al., 2017; Lin et al., 2015; Noziére et al., 2015; De Haan et al., 2018; Hawkins et al., 2018), but more studies are necessary to accurately determine the reactions that occur and the products formed within the atmospheric aqueous phase. SFC is a useful tool for the separation of these reaction mixtures and can add to current analysis techniques to provide more information about the reaction products formed.

*Author contributions.* MMG designed the experiments and DNG, MBS, and MMG carried them out. MMG prepared the manuscript with contributions from all co-authors.

*Competing interests.* The authors declare that they have no conflict of interest.

*Acknowledgements.* The authors thank Dr. Lindsay Soh and Dr. Joseph Woo for helpful discussions about chromatography and mass spec-

5  trometry. We also thank Jaqueline Sharp for manuscript comments. This work was funded by the National Science Foundation (MRI-1626100).

*Data availability.* Chromatograms are publicly available as text files at http://sites.lafayette.edu/gallowam/publications/ (last access: 31 May 2019).

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
