# Peer review of "Separation and detection of aqueous atmospheric aerosol mimics using supercritical fluid chromatography–mass spectrometry"

_Atmospheric Measurement Techniques, 2019_

## Referee Comment (RC1) · Anonymous Referee #1 · 14 Mar 2019

- GENERAL REMARKS The authors present a supercritical fluid chromatography–mass spectrometry method for separation and detection of aqueous atmospheric aerosol mimics. In this study SFC-MS was used to study methylglyoxal and ammonium sulphate creation mixture as mimics of reaction mixtures in atmospheric droplets. ESI and APCI ionisation modes were used for the detection of various species present in the reaction mixture. five different columns were screened to optimise separation and fourteen reaction products, detected for the first time, were reported. The study address challenges like separation of compounds with different polarities and reduction of analysis time. Identification of unknown fragments/compounds can be a strength of the work presented here. The study is relevant for the scientific community however

the study design is not comprehensive and several important aspects of experimental work are not completely described. I give some suggestions hereinafter.

- MAJOR COMMENTS - Aerosols are a mixture of solid particles and liquid droplets suspended in gases (air). The terms use of terms e.g. aqueous molecules, aqueous atmospheric systems and atmospheric droplets should be explained and the terminology should be consistent throughout the text to assist readers. - Introduction needs to be revised, ideally introduction should address 1) gaps in knowledge, 2) specific research question(s), 3) approach used to answer the research question(s) and 4) comparison with already available knowledge. In the current state, large part of introduction focuses on the theory of SFC which better fits in an SFC (P2, L28-35 and P3, L1-6 needs to be revised and should focus more on the analytes in question). - Authors compare SFC with LC and GC. With the development of UHPLC, analysis time has significantly reduced. Describing the benefits of SFC should not be stop having a nice comparison with available UHPLC methods. - In modern SFC, there is a huge range of packed columns available today. The authors should motivate why BEH (three types), HILIC and C18 columns were used for the screening for suitable stationary phase. - Section 2.3.2, L9 (optimal mobile phase conditions varied slightly with the identity of the column........). Why different mobile phase conditions were used to compared column efficiencies? For any comparison all the variables must be same except the one subject to comparison. Secondly what were the varied mobile phase compositions used for comparison? Why not to make use of supplementary information and add a figure/table to describe the actual experimental conditions? - Section 3.2.2, L6-7: include chromatograms in supplementary information - Section 3.3, L30-34: include mass spectra in supplementary information - ESI and APCI methods were not optimised for higher signal of the analyte, therefore, it is inappropriate to claim that APCI is not a better method based on the results. However, more information can be included from literature to motivate if APCI is a suitable ionisation source for polar compounds.

- MINOR COMMENTS - Suitable keywords should be included with abstract - P2, L3-4:

[Figure]

include a reference - Section 2.2, L10: "the mixture was allowed to react for at least a month......"; an accurate time must be included - P4, L22: add "that" between "to ensure" and "the mobile phase" - P6, L25: "......polar molecular interactions between analytes may be driving the solution through the column", a reference must be added to support the assumption - P7, L5: "........although all temperatures and modified conditions discussed below were tested on each column with similar results"; it is insufficient to state "similar results" when there is a possibility to include chromatograms in supplementary information and generate a more quality discussion - P7, L16: Its better to discuss the strengths/weaknesses of certain mobile phase in relation to properties of analytes rather then SFC itself. - P9, L8-15: the text should be revised considering both mobile phase density and kinetic effects should be considered in relation to retention times - The language needs revision in terms of use of article "the"

---

## Referee Comment (RC2) · Anonymous Referee #2 · 15 Mar 2019

**GENERAL COMMENTS**

The authors report on the development of supercritical fluid chromatography for the separation of polar products of the atmospherically important reaction between methylglyoxal and ammonium sulfate. The reaction itself has already been widely investigated. New molecular/fragment ions were found, however identification of the corresponding analytes was not in focus of the study (is also not expected due to the unit mass resolution of mass spectrometric detection). The presented SFC is an attractive and greener alternative to commonly applied LC and GC methods, but the motivation why it was developed for the analysis of the investigated reaction is not clear. Also, its better performance in comparison to conventional analytical techniques is not well justified (see below). Moreover, the use of C18 and HILIC columns seems fundamentally inappropriate; one does not expect any good results when applying nonpolar-to-polar gradient on C18 or operating HILIC without a certain amount of water.

It should be made clear, by corrections throughout the manuscript, that there is no chromatographic method that is unique and can be used for the detection of any analyte in any mixture. In this regard, it should be clearly shown at the end of the manuscript why the new chromatographic method is better performing than the conventional LC/GC separations (best by comparison of SFC, LC and GC chromatograms, a real sample analysis would be above expectations). I believe that the new identified peaks cannot be unambiguously attributed to the better separation, but may also arise from different MS detection (different instrument/ESI source, lower LOD, etc.). Please revise the manuscript addressing these issues in particular.

**SPECIFIC COMMENTS**

P1L3: *These methods (GC and LC) can be time-consuming and do not easily separate highly polar aqueous molecules.* -> The presented method obviously also doesn't assure separation of highly polar products (broad peak after 11 min).

P2L13-17: First, use of ion-pairing reagents enables/improves separation of polar analytes on RP columns and has for instance been successfully applied to the detection of ambient organosulfates. Second, how long the method has to be is very much dependent on the complexity of the sample (simulated reactions are usually less demanding than real aerosol extracts). Thirdly, many peaks co-elute also in your case (broad peak after 11 min).

P2L35-P3L1: not strictly true, revise

Section 2: a summary (table) of all tested conditions is missing (best to put it in SI).

2.3.1: four different BEH columns were used and only one is shortly named BEH. This may be misleading. I suggest changing this acronym.

P6L11 and Fig.2: Amide column does not seem any better than C18 and HILIC – improve data representation or revise the text.

P6L13-14: how do you know how many compounds elute after 12 min? It is better to say that most compounds efficiently separate within 12 min...

P6L20-29: As already stated above, the usage of C18 and HILIC seems fundamentally inappropriate. If they were treated differently, explain in detail how.

P6L26: BEH Amide and 2-EP are not HILIC columns, but rather contain polar stationary phase.

P8L1: I don't understand: *elute much more cleanly from the column*

P8L6: the reaction was left for 1 month to get sufficient amounts of products for the detection, so I don't expect that a few minutes of reaction between the carbonyls and ammonium on the column can produce the measured artefacts.

P8L16-17: same also for LC and GC

P10L29: the newly identified low-intensity signals are not always separated on the column (see for instance m/z 83,87,98,139 etc.) – they probably appear because of better performing MS detection. Also, when EIC is measured, the quality of chromatographic separation often doesn't need to be supreme; selectivity is already assured with the selection of the ion.

---

## Author Comment (AC1) · 30 May 2019

*We thank the reviewer for their included suggestions, questions, and points for clarification. We address the reviewer's feedback below. Our responses to the reviewers are included in italics after each reviewer comment. Additionally, the revised version of the manuscript is added.*

-GENERAL REMARKS The authors present a supercritical fluid chromatography–mass spectrometry method for separation and detection of aqueous atmospheric aerosol mimics. In this study SFC-MS was used to study methylglyoxal and ammonium sulphate creation mixture as mimics of reaction mixtures in atmospheric droplets.

[Figure]

ESI and APCI ionisation modes were used for the detection of various species present in the reaction mixture. five different columns were screened to optimise separation and fourteen reaction products, detected for the first time, were reported. The study address challenges like separation of compounds with different polarities and reduction of analysis time. Identification of unknown fragments/compounds can be a strength of the work presented here. The study is relevant for the scientific community however the study design is not comprehensive and several important aspects of experimental work are not completely described. I give some suggestions hereinafter.

- MAJOR COMMENTS

- Aerosols are a mixture of solid particles and liquid droplets suspended in gases (air). The terms use of terms e.g. aqueous molecules, aqueous atmospheric systems and atmospheric droplets should be explained and the terminology should be consistent throughout the text to assist readers.

*We have revised the manuscript with respect to this suggestion, and have made our terminology more consistent.*

- Introduction needs to be revised, ideally introduction should address 1) gaps in knowledge, 2) specific research question(s), 3) approach used to answer the research question(s) and 4) comparison with already available knowledge. In the current state, large part of introduction focuses on the theory of SFC which better fits in an SFC (P2, L28-35 and P3, L1-6 needs to be revised and should focus more on the analytes in question).

*We have revised and restructured major parts of the introduction and removed some of the general SFC theory while focusing more on the atmospheric compounds in question.*

-Authors compare SFC with LC and GC. With the development of UHPLC, analysis time has significantly reduced. Describing the benefits of SFC should not be stop having a

nice comparison with available UHPLC methods.

*While revising the introduction to address the concerns above, we have added a discussion of current UPLC methods in use for this and similar systems.*

[revised manuscript text omitted]

-Section 2.3.2, L9 (optimal mobile phase conditions varied slightly with the identity of the column........). Why different mobile phase conditions were used to compared column efficiencies? For any comparison all the variables must be same except the one subject to comparison.

*We agree that with a true comparison, only one factor should change. However, we did not intend to compare all columns with the same mobile phase composition. Instead, our intention was to compare the best chromatography we are able to achieve from each column to determine which columns might be most useful for these analyses. However, due to the nature of these systems, the optimized mobile phase conditions were very similar, as can be seen in the newly created Table S2 in the Supplemental Information. Most of the differences in gradient profiles were due to a slightly different ramping speed for the modifier.*

Secondly what were the varied mobile phase compositions used for comparison? Why not to make use of supplementary information and add a figure/table to describe the actual experimental conditions?

*We have taken the advice of both referees and added Table S2 to the Supplemental Information with this information.*

- Section 3.2.2, L6-7: include chromatograms in supplementary information

*We have added several figures (Figs. S1-9) to the supplemental information, including those showing all the chromatograms taken for the acetonitrile, methanol, and methanol with formic acid modifiers as well as a comparison between similar modi-*

*fiers on each column. These show the same data, but help to show how both columns and modifiers affect separations.*

- Section 3.3, L30-34: include mass spectra in supplementary information

*We have added figures S12-S14 to the supplementary information showing the fragmentation data for the peaks we discuss in this section.*

- ESI and APCI methods were not optimized for higher signal of the analyte, therefore, it is inappropriate to claim that APCI is not a better method based on the results. However, more information can be included from literature to motivate if APCI is a suitable ionisation source for polar compounds.

*While it is time consuming to optimize either ESI or APCI conditions for each compound in such a complex mixture, optimization was performed with both ionization methods to ensure a large range of compounds were detected with each method. We have made this clear in the text:*

*"Data collection was performed with both ESI and APCI ionization modes for comparative purposes. Optimizing the MS method for each individual mass is time-consuming during both method development and data collection, so each ionization method was generally optimized to maximize as many signals as possible using direct infusion into the MS. The mobile phase in SFC separations is acidic (Lesellier and West, 2015), which helps to protonate analyte molecules during the ESI ionization process. Therefore, it is not surprising that the overwhelming majority of these compounds are detected in positive ESI mode since they contain alcohols and nitrogen-containing functional groups that are easily protonated. The presence of formic acid in the makeup flow (ESI solvent) enhances ionization of any compound more basic than the resulting solution, making its addition very useful for these analyses. Most compounds were also detected in APCI mode, but at much lower intensities than in ESI mode ($\sim 20\times$, see Fig. S12). Therefore, ESI is the preferred mode for analysis of this aerosol mimic system, though there may be some compounds that have lower ionization efficiency*

*with ESI and may benefit from the use of APCI in some solutions, as it has often been used for analysis of slightly less polar compounds such as polycyclic aromatic hydrocarbons, esters, and pyrazine derivatives (Walgraeve et al., 2010; Laskin et al., 2015; Noziére et al., 2015; Laskin et al., 2017; Hawkins et al., 2018). As a range of polarities are found in many atmospheric samples, the ability to switch back and forth between modes in the same separation is useful for such analyses. All masses observed in this study are given in Table S3 and all extracted ion chromatograms (EICs) are shown in Fig. S14. The chromatograms presented in this study are a combination of all EIC signals in Table S3 and Fig. S14."*

MINOR COMMENTS

- Suitable keywords should be included with abstract

*Keywords are not included for AMT manucripts.*

- P2, L3-4: include a reference

*Thank you, references have been added to these lines. The text now reads:*

*"The aqueous aldehyde – ammonium/amine reaction mixture is one class of reaction systems that has been shown to impact aerosol growth (Lin et al., 2015; Hawkins et al., 2016; Aiona et al., 2017; De Haan et al., 2017)."*

- Section 2.2, L10: "the mixture was allowed to react for at least a month......"; an accurate time must be included

*We have included more specific information here, and the text now reads:*

*"Due to slow room temperature reaction times at these concentrations, the mixture was allowed to react for 6-7 weeks in a dark environment before analysis (Zhao et al., 2015)."*

- P4, L22: add "that" between "to ensure" and "the mobile phase"

*We have changed this wording.*

- P6, L25: "......polar molecular interactions between analytes may be driving the solution through the column", a reference must be added to support the assumption

*During our revision of the discussion of stationary phases, this line was removed from the text.*

- P7, L5: "........although all temperatures and modified conditions discussed below were tested on each column with similar results"; it is insufficient to state "similar results" when there is a possibility to include chromatograms in supplementary information and generate a more quality discussion

*We have added several chromatograms to the Supplemental Information to help us expand upon this discussion and referred our readers to these figures in the text for clarification.*

- P7, L16: Its better to discuss the strengths/weaknesses of certain mobile phase in relation to properties of analytes rather then SFC itself.

*Generally, we agree that discussing the mobile phase with respect to the analyte is more useful than blanket statements about the use of additives in a chromatography method. However, we are trying to make the point here that ammonium formate (or small amine salts) are often used as mobile phase additives in SFC and this is why we tested ammonium formate as an additive. We then go on to describe the drawbacks of using such a modifier in a system containing carbonyl analytes. This was not immediately obvious to us, and indeed our other referee points out that "I don't expect that a few minutes of reaction between the carbonyls and ammonium on the column can produce the measured artefacts." Therefore, we feel that it is important to make the point that while ammonium formate is a common SFC additive, we must be careful of the reactions that occur within the column or ionization source, even if we do not expect them to occur.*

- P9, L8-15: the text should be revised considering both mobile phase density and kinetic effects should be considered in relation to retention times

*We have revised the text to discuss how solvating power changes with temperature and density, and expanded our discussion of both the theory behind the temperature effects and the results of such effects, putting more focus on the results. This is a large section of text, so is not given immediately below, but can be found in the revised manuscript.*

- The language needs revision in terms of use of article "the"

*As we have revised the text, we have done so with this in mind.*

REFERENCES:

[revised manuscript text omitted]

Please also note the supplement to this comment:
https://www.atmos-meas-tech-discuss.net/amt-2019-42/amt-2019-42-AC1-supplement.pdf
* * *

---

## Author Comment (AC2) · 30 May 2019

*We thank the reviewer for their included suggestions, questions, and points for clarification. We address the reviewer's feedback below. Our responses to the reviewers are included in italics after each reviewer comment. Additionally, the revised version of the manuscript is added.*

GENERAL COMMENTS The authors report on the development of supercritical fluid chromatography for the separation of polar products of the atmospherically important reaction between methylglyoxal and ammonium sulfate. The reaction itself has already been widely investigated. New molecular/fragment ions were found, however identifi-

cation of the corresponding analytes was not in focus of the study (is also not expected due to the unit mass resolution of mass spectrometric detection). The presented SFC is an attractive and greener alternative to commonly applied LC and GC methods, but the motivation why it was developed for the analysis of the investigated reaction is not clear. Also, its better performance in comparison to conventional analytical techniques is not well justified (see below). Moreover, the use of C18 and HILIC columns seems fundamentally inappropriate; one does not expect any good results when applying nonpolar-to-polar gradient on C18 or operating HILIC without a certain amount of water.

It should be made clear, by corrections throughout the manuscript, that there is no chromatographic method that is unique and can be used for the detection of any analyte in any mixture. In this regard, it should be clearly shown at the end of the manuscript why the new chromatographic method is better performing than the conventional LC/GC separations (best by comparison of SFC, LC and GC chromatograms, a real sample analysis would be above expectations). I believe that the new identified peaks cannot be unambiguously attributed to the better separation, but may also arise from different MS detection (different instrument/ESI source, lower LOD, etc.). Please revise the manuscript addressing these issues in particular.

*We agree that no chromatographic method is perfect for every analyte, and have tried to make it clearer throughout the manuscript that SFC is an alternative to LC that may be able to provide complementary information on these chemical systems, but is not necessarily the only tool that will provide useful information. We have also revised much of the text with a discussion of how it is the combination of the separation of compounds such as isomers and the use of EIC and tandem MS that make it possible to learn more about this methylglyoxal/AS system. In addition, it is likely that a lower LOD is an advantage here, but without knowing enough details about the specific ESI-MS and APCI-MS instruments used in past studies, we are not comfortable making this claim.*

SPECIFIC COMMENTS

P1L3: These methods (GC and LC) can be time-consuming and do not easily separate highly polar aqueous molecules. -> The presented method obviously also doesn't assure separation of highly polar products (broad peak after 11 min).

*We agree that our separation method does not assure separation of all compounds, but we think this strengthens our point that separation is difficult with these systems. We have revised the abstract and multiple sections throughout the text to make it clear that while complete separation does not occur with the presented chromatography, the use of EIC and MS/MS still allows for analysis of these coeluting compounds.*

P2L13-17: First, use of ion-pairing reagents enables/improves separation of polar analytes on RP columns and has for instance been successfully applied to the detection of ambient organosulfates. Second, how long the method has to be is very much dependent on the complexity of the sample (simulated reactions are usually less demanding than real aerosol extracts). Thirdly, many peaks co-elute also in your case (broad peak after 11 min).

*We have revised this section of the introduction to talk in general about the use of GC and LC for atmospheric system, and then we discuss more specifically the few studies that have used LC to study reactions of small carbonyls with amines or ammonium. Due to this revision of the text, we do not feel that the mention of ion pairing fits into the manuscript because, to our knowledge, ion pairing reagents are not often used for this particular system. However, we do mention that the additives we are using are ion-pairing agents. With the use of EIC and MS/MS, the complete separation of these peaks is not completely necessary to identify and study these compounds, and previous studies have not achieved complete separation, but were still able to draw many conclusions about this system. This technique is complementary to previous techniques. As we revised the text based on other referee comments, we made many changes to how we discuss this method in comparison to others, and believe we have*

*made this point much more clearly.*

*"A significant challenge to the identification and quantification of atmospheric reaction mixtures is the separation of the compounds that compose them. This is due in part to the high degree of similarity between many of the compounds in solution (Noziére et al., 2015). The two most commonly used separation techniques are gas chromatography (GC) and liquid chromatography (LC). For many atmospheric samples, derivatization must be performed before GC analysis, which leads to increased specificity and identification. However, derivatization can also lead to side reactions and ambiguity in structural identification (Noziére et al., 2015). Reverse-phase high performance LC (HPLC) and ultra performance LC (UPLC) are also commonly coupled to mass spectrometry (MS) for separation and identification of atmospheric compounds (Lin et al., 2015; Noziére et al., 2015; Aiona et al., 2017; De Haan et al., 2018; Jayarathne et al., 2018), and several studies have used these techniques to study aldehyde – ammonium/amine reaction systems (Lin et al., 2015; Aiona et al., 2017; Kampf et al., 2016; Kampf et al., 2012). Lin et al. (2015) and Aiona et al. (2017) provided comprehensive studies of chromophores found in the methylglyoxal – ammonium sulfate system before and after photolysis using similar HPLC methods. Lin et al. (2015) found that an acetonitrile/water gradient with an SM-C18 column provided the best separation in 80 minutes at 0.2 mL min$^{-1}$. Kampf et al. (2012) analyzed a glyoxal – ammonium sulfate mixture with an acetonitrile/water gradient on an Atlantis T3 (C18) column in 60 minutes at 0.2 mL min$^{-1}$. A similar study analyzed the nitrogen-containing compounds from the reaction of small dicarbonyls and amines on HPLC and UPLC (Kampf et al., 2016). The HPLC method utilized the same Atlantis T3 column and an acetonitrile/water gradient to separate these reaction mixtures in 19 minutes at 0.5 mL min$^{-1}$, while the UPLC method used a Hypersil Gold C18 column with an acetonitrile/water with formic acid gradient to separate the compounds in 8.5 minutes at 0.5 mL min$^{-1}$ for analysis via targeted MS/MS. The use of tandem MS coupled to both chromatography systems in that study allowed for the identification of many compounds without complete separation. While GC and LC have provided many important insights into*

*numerous atmospheric systems, there is a need for a separation method for aldehyde – amine reaction systems that does not require derivatization and can reduce the necessary separation time while still providing separation of a majority of the compounds in the mixture."*

P2L35-P3L1: not strictly true, revise

*We have altered this text to be consistent with our original intended meaning, which is that modifiers and additives can expand the polarity range that can be achieved in a single chromatographic run compared to that of pure carbon dioxide mobile phase. The text now reads:*

*"Incorporating modifiers and additives can change the polarity of the mobile phase, thereby making it possible to separate a range of polar or nonpolar compounds and allowing SFC to be used analogously to either normal- or reverse-phase LC (Guiochon and Tarafder, 2011). As it is possible to use a mobile phase gradient that ranges from pure carbon dioxide to ∼50% modifier, the mobile phase can be changed from nonpolar to relatively polar over the course of one injection onto the column. This makes SFC ideal for the separation of mixtures containing compounds with a wide range of polarities, such as the methylglyoxal – ammonium sulfate reaction system."*

Section 2: a summary (table) of all tested conditions is missing (best to put it in SI).

*We have taken the advice of both referees and added Table S2 to the supplemental information showing the chromatography conditions.*

2.3.1: four different BEH columns were used and only one is shortly named BEH. This may be misleading. I suggest changing this acronym.

*We have clarified this by changing the abbreviation for all of the BEH columns to include "BEH" in the abbreviation.*

P6L11 and Fig.2: Amide column does not seem any better than C18 and HILIC – improve data representation or revise the text.

*We have remade the figures to make the early elution off the columns clearer and added more figures to the supplemental information. We have also updated the text slightly, and it now reads:*

*"It was not possible to separate the compounds of interest with the BEH C18 column, as can be seen in Figs. 2 and S1-S8, likely due to the nonpolar stationary phase. With a methanol modifier, most compounds eluted within 2 minutes even when the starting conditions contained 100% carbon dioxide. Varying degrees of separation were observed with the HILIC-based stationary phases (HILIC, BEH Amide, BEH 2-EP, and BEH). Elution times for many compounds range from <1 minute to 20 minutes, which is approximately when the mobile phase reaches its most polar condition. EICs for the first 11 minutes on each column using a methanol modifier are shown in Fig. 2, and all other chromatograms are shown in Figs. S1-S8. It was expected that the HILIC column would efficiently separate this mixture, since it is composed of bare silica and is often used in separations of similar, highly polar aqueous mixtures (Laskin et al., 2017). However, while separation on the HILIC column was improved over that of the BEH C18 column, there were still many wide, coeluting peaks between 1 and 3 minutes along with peaks that eluted much later (7 and 13 minutes), indicating that while some separation is occurring, many compounds are not well separated. BEH columns with polar functionalities such as the BEH Amide and BEH 2-EP combined with a polar methanol modifier provided improved separation, as the methylglyoxal – ammonium sulfate mixture produces many polar compounds that contain nitrogen- and oxygen-containing functional groups that have heightened interactions with the nitrogen-containing stationary phase (amide or 2-ethylpyridine) in these columns. However, both exhibited similar features as the HILIC column, with several compounds eluting within 3 minutes, followed by wider peaks between 4 and 9 minutes."*

P6L13-14: how do you know how many compounds elute after 12 min? It is better to say that most compounds efficiently separate within 12 min... We agree that this clarification will make the text more accurate, and we have changed the wording to

state that:

*"All columns provided better separation of the compounds that eluted within 11 minutes than later eluting compounds (e.g., m/z 83, 97, and 126), which either coeluted or had very wide, noisy peaks that overlapped significantly depending on eluent conditions (see Figs. S1-S8).*

P6L20-29: As already stated above, the usage of C18 and HILIC seems fundamentally inappropriate. If they were treated differently, explain in detail how.

*We disagree that the BEH C18 and HILIC columns are fundamentally inappropriate for SFC separations. There is literature precedent for using both types of columns with this chromatography system. C18 columns have been used with SFC in the past to separate mixtures of imidazole derivatives, including some of the same compounds we are analyzing herein (Patel et al., 1998; Parlier et al., 1991). We found that the BEH C18 column could not separate the compounds of interest very well, but there was minimal separation. Due to the reverse-phase nature of this column, we also initially tested a polar-to-nonpolar gradient and found that there was no separation of the peaks, indicating that while this column may not work well for this system, there some separation does occur when used in a nonpolar-to-polar solvent gradient.*

*HILIC columns have been successfully used with SFC using a variety of polar organic solvents as a modifier (methanol, acetonitrile, and isopropanol with small acids and amine additives) (Bieber et al., 2017; Dispas et al., 2012; Lesellier and West, 2015; West et al., 2012). While water is useful for these separations, separation can still be achieved with the CO2/polar organic solvent gradients used for SFC. In fact, several of the columns used herein are HILIC columns (HILIC, BEH, BEH Amide, and BEH 2-EP) and all provide varying degrees of separation between compounds. The HILIC column used here did not provide the same chromatographic resolution as the BEH column, but was able to separate many of the compounds in the mixture.*

P6L26: BEH Amide and 2-EP are not HILIC columns, but rather contain polar station-

ary phase.

*While these BEH columns do contain a polar stationary phase, they are based on HILIC technologies and are often considered to be HILIC columns by the manufacturer (Waters) and according to several published studies (Kitanovski et al., 2012; King et al., 2019). The column chemistry is slightly different from a standard HILIC column, but has many similarities, as are now discussed in the text.*

*"The HILIC column is a polar unbonded stationary phase and the BEH stationary phase is an ethylene bridged HILIC formulation. Both are intended to separate polar compounds. The BEH Amide and BEH 2-EP columns are modified BEH columns, with amide or 2-ethylpyridine groups bonded to the stationary phase. Both columns have previously been used for the separation of polar compounds containing amines and alcohols, functional groups found in the methylglyoxal – ammonium sulfate reaction mixture (Lesellier and West, 2015)."*

P8L1: I don't understand: elute much more cleanly from the column

*Thank you for pointing out this unclear wording. We have changed the text to explain the differences we see in the chromatogram with the ammonium formate additive. The text now reads:*

*"When using 10 mM ammonium formate in methanol as the mobile phase modifier, separation of compounds that elute in less than 11 minutes is similar to that of a pure methanol modifier (Fig. 3), and the compounds that elute after 11 minutes do so with better resolution and with sharper peaks than the methanol or methanol with formic acid modifiers."*

P8L6: the reaction was left for 1 month to get sufficient amounts of products for the detection, so I don't expect that a few minutes of reaction between the carbonyls and ammonium on the column can produce the measured artefacts.

*We agree that the interaction of carbonyl analytes and ammonium on the column is not*

*likely to have caused the increase in nitrogen-containing products, rather that the rapid drying of the droplets in the ESI source is where this reaction is occurring. However, we have not tested this hypothesis and therefore only know that an interaction between the species is occurring somewhere in the SFC-MS system. We have changed the wording in the manuscript to state that:*

*"However, the addition of ammonium in the mobile phase or makeup flow leads to artificially high signals from nitrogen-containing compounds, even in samples that contain no nitrogen (e.g., aqueous methylglyoxal). These compounds are also seen in samples containing ammonium sulfate, but their signals are enhanced with additional ammonium added into the system via the mobile phase or makeup flow and do not always elute at the same times as in these systems. Thus, the increased nitrogen-containing compounds are being formed within the instrument. Ammonium is reacting with carbonyls in the sample either on the column or in the ionization source, most likely in the ionization source. Previous studies have noted increased oligomer signals as a result of ESI ionization, likely due to the rapid increase in analyte concentrations within the droplets upon drying (Hastings et al., 2005). This is likely happening here, with methylglyoxal and ammonium reacting within the droplets. This is further supported by the fact that while some earlier masses elute at similar times to the methanol system, there are nitrogen-containing compounds detected at times that do not match peaks eluted with a pure methanol modifier. It is possible that compounds eluting from the column at this time are methylglyoxal oligomers formed through aldol condensation that then react with the ammonium after elution (Krizner et al., 2009). It is also possible for analytes to react with the mobile phase during SFC analysis (Lesellier and West, 2015)."*

P8L16-17: same also for LC and GC

*While it is true that temperature changes the mobile phase in all chromatographic systems discussed, the effects of temperature on SFC can be counterintuitive to those used to thinking in terms of the temperature effect on viscosity and therefore retention*

*time in GC or LC. We have changed the text to make this difference clearer, and it now reads:*

*"The solvating power of a super- or sub-critical fluid depends on the density of the fluid, which is affected by the temperature and pressure of the system. Therefore, SFC is similar to GC and LC in that retention and separation are not only controlled by the stationary and mobile phases, but also temperature and pressure (Saito, 2013). Column temperature may become an important parameter for separation of compounds and can significantly change retention. At lower temperatures, mobile phase density and solvating power increase. When this occurs, retention times tend to shorten, which is the opposite of what would be expected for GC or LC (Saito, 2013). However, this is balanced by the effect of lower temperatures on the kinetic partitioning of analytes into the stationary phase as is typically seen in LC or GC. Due to these several convoluting factors affecting analyte-mobile phase interactions, it is useful to test the effect of temperature on the separation of these mixtures."*

P10L29: the newly identified low-intensity signals are not always separated on the column (see for instance m/z 83,87,98,139 etc.) – they probably appear because of better performing MS detection. Also, when EIC is measured, the quality of chromatographic separation often doesn't need to be supreme; selectivity is already assured with the selection of the ion.

*The reviewer is correct, and there is a combination of factors that leads to the detection of these new masses. We have reworded this text to make it clearer that the separation method is not the only factor in play here and made changes throughout the document to make it clear that there are many factors that allow us to see these compounds.*

*"While it is not surprising to detect methylimidazole compounds within this system, the combination of tandem MS and chromatography that allows for separation of compounds with similar masses allows for the observation of these low intensity signals that have not been identified in previous studies."*

Many other small aldehydes and organic acids are found within  aqueous SOA that react with ammonium and other amines to form similar product mixtures to that studied here. These have been explored by several groups (Heald et al., 2005; Hallquist et al., 2009; Jimenez et al., 2009; Laskin et al., 2017; Lin et al., 2015; Noziére et al., 2015; De Haan et al., 2018; Hawkins et al., 2018), but more studies are necessary to accurately determine the reactions that occur and the products formed within the atmospheric aqueous phase. SFC is a useful tool for the separation of these reaction mixtures and can  add to current analysis techniques to provide more information about the reaction products formed.

*Author contributions.* MMG designed the experiments and DNG, MBS, and MMG carried them out. MMG prepared the manuscript with contributions from all co-authors.

*Competing interests.* The authors declare that they have no conflict of interest.

*Acknowledgements.* The authors thank Dr. Lindsay Soh and Dr. Joseph Woo for helpful discussions about chromatography and mass spectrometry. We also thank Jaqueline Sharp for manuscript comments. This work was funded by the National Science Foundation (MRI-1626100).

*Data availability.* Chromatograms are publicly available as text files at http://sites.lafayette.edu/gallowam/publications/ (last access: 31 May 2019).

25